# Water Use Strategies of Dominant Species (*Caragana korshinskii* and *Reaumuria soongorica*) in Natural Shrubs Based on Stable Isotopes in the Loess Hill, China

Yu Zhang [1], Mingjun Zhang [1,*], Deye Qu [1], Wenguang Duan [2], Jiaxin Wang [1], Pengyan Su [1] and Rong Guo [1]

[1] College of Geography and Environmental Science, Northwest Normal University, Lanzhou 730070, China; 2018212277@nwnu.edu.cn (Y.Z.); qudeye@nwnu.edu.cn (D.Q.); 2019212377@nwnu.edu.cn (J.W.); 2018212291@nwnu.edu.cn (P.S.); 2017212149@nwnu.edu.cn (R.G.)
[2] Lanzhou Meteorologic Bureau, Lanzhou 730020, China; maqian0516@nwnu.edu.cn
* Correspondence: mjzhang@nwnu.edu.cn

**Abstract:** Water is a key and limiting factor for ecosystem processes (carbon dioxide fixation, vegetation growth, respiration, etc.) and functions (NPP, Net Primary Productivity) in arid and semi-arid areas. As the main regulator, knowledge of plant water use patterns is essential in understanding the SPAC (Soil-Plant-Atmosphere-Continuum) cycle. The tree-planting project in the southern and northern mountains of Lanzhou city aims to improve the ecological environment and promote urban construction. In this study, we analyzed the water use strategies of the dominant plants *C. korshinskii* and *R. soongorica* in natural shrubs of southern and northern mountains of Lanzhou city using oxygen stable isotope techniques. The result showed that the flexible water uptake pattern of *C. korshinskii* and its faster response to precipitation pulse, compared with *R. soongorica*, might help it to make full use of water and nutrients and adapt to the dry environment. However, *R. soongorica* progressively switched to suck up deeper soil water and increased the water use proportion from 0.5% to 84.4% as the seasons changed, indicating a greater degree of ecological plasticity. The flexible water use strategies of *C. korshinskii* and *R. soongorica* in the same habitat reduced competition for water and nutrients and enhanced adaptability to arid environments. The work presented here provides insights into vegetation restoration and ecological management for the southern and northern mountains of Lanzhou city.

**Keywords:** natural shrubs; water use strategy; MixSIAR; stable isotopes

## 1. Introduction

Water is a crucial factor restricting the growth and development of plants in arid and semi-arid areas and determines the distribution and ecological functions of vegetation. Plant water use plays an important role in understanding and simulating hydrological processes at the soil–vegetation–atmosphere interface [1–4], reflects the adaptability of plants to the environment and provides important insights into the adaptability of plants to the changing environments [5–7]. *Caragana korshinskii* is leguminous *Caragana* genus plant, which is an extreme xerophytic shrub. It mainly grows in hills, mountain, and mountain ravines or in mixed mountain shrub areas [8]. What is more, it has flourishing roots and high adaptability and is a preferred shrub for the conservation of water and soil and ecological restoration in arid semi-arid area. *Reaumuria soongorica* is an extreme xerophytic dwarf shrubs of the Tamaricaceae family and *Reaumuria* genus, belongs to the cretaceous

and is a relic plant of the tertiary period [9]. It mainly grows in mountain hills, Gobi, desert and the semi-desert piedmont plain [10]. Furthermore, it is one of the dominant and constructive species in the temperate desert steppe and desert steppe areas in China. As a zonal type of vegetation in the arid loess plateau, it plays an important role in vegetation restoration and reestablishment and soil erosion control. Recent studies showed that the precipitation in the northwest has increased significantly, with the temperature substantially increasing, and the potential evaporation has grown sharply for the past few years [11], which will cause changes in its hydrological cycle.

Water sources utilized by vegetation can be determined by many approaches, such as root system excavation [12], sap flow techniques [13], electrical resistivity [14], GIS tools [15] and the radioactive tracer tritium [16]. Although these methods could determine the water source of plant to a certain extent, the stable isotope technique provides an accurate, effective and efficient and less destructive approach for evaluating and identifying the water use strategies of vegetation. Previous studies have proved that isotopic fractionation does not occur in the process of plant root absorption and transportation of water [5,17,18]. However, Ellsworth and Williams [19] found that the roots of xerophytes and halophytes undergo fractionation of hydrogen isotopes when they absorb water [20], if only oxygen isotopes were used in this study. Therefore, it is possible to determine whether the water sources of plants comes from superficial or deeper soil [13], precipitation or fog [21], runoff or soil water [22].

Phillips and Gregg [23] proposed a multilinear compound model, IsoSource (https://www.epa.gov/eco-research/stable-isotope-mixing-models-estimating-source-proportions), based on the law of conservation of isotope mass, which has been widely applied. However, it has disadvantages, such as poor precision and lack of consideration of the analysis of uncertainty. Considering that, Moore [24], Parnell [25] and Stock [26] proposed Bayesian isotope mixing models sequentially (MixSIR, SIAR, MixSIAR), which fuse uncertainties associated with multiple sources, discrimination factors and the uncertainty of isotopic data. In addition, it can combine prior information to make the model estimation more precise and accurate [27–29]. Wang et al. [30] compared three Bayesian mixed models (MixSIR, SIAR and MixSIAR) and reported that the SIAR and MixSIAR models have better water source apportionment performances than the MixSIR model. Therefore, we used the MixSIAR model to analyze the water sources of plants.

The southern and northern mountains of Lanzhou city are located in a typical semi-arid area with low forest and grass coverage, sparse vegetation and a fragile ecological environment. In addition, the environmental condition varies from natural forests in the southern alpine areas to large areas of loess hills in the middle and desert grasslands in the north [31–33]. The green project in the southern and northern mountains is of great significance in improving the quality of the ecological environment in Lanzhou and promoting urban construction. The current research mainly focuses on vegetation restoration [34], vegetation types [33], community structure and plant diversity surveys [35,36], but the water use patterns of typical plants remain poorly understood in this region. Natural vegetation is the main regulator of the SPAC (Soil-Plant-Atmosphere-Continuum) cycle [37,38]. Studying the water sources of the dominant shrubs *C. korshinskii* and *R. soongorica* in the natural shrubs of the southern and northern mountains of Lanzhou city, and exploring the water use strategies and adaptation mechanisms of the plants in the context of global warming are important for assessing the ecological sustainable development of the region under the climate change scenario.

In this study, we analyzed the water use strategies of the dominant plants *C. korshinskii* and *R. soongorica* in natural shrubs of the southern and northern mountains of Lanzhou city using oxygen stable isotope techniques (MixSIAR model). Furthermore, we discussed water adaptation mechanisms of natural shrubs in the southern and northern mountains of Lanzhou city by analyzing seasonal changes of water absorption depth and the transformation of plant water use patterns after rainfall, which provides a decision making basis for the ecological restoration and effective utilization of plant resources in the green engineering of the southern and northern mountains of Lanzhou city.

The purposes of this paper were to: (i) investigate the isotopic compositions of soil water and their vertical gradients along the soil profile, and (ii) quantify the seasonal variations in water use strategies and determine their differences among the two natural shrubs (*C. korshinskii* and *R. soongorica*).

## 2. Materials and Methods

### 2.1. Study Area

Lanzhou city (35°34′–37°07′ N, 102°35′–104°34′ E) is located in the northwest of China and the central Gansu Province. It belongs to the transition zone between the eastern humid monsoon climate zone, the Qinghai–Tibet Plateau alpine region and the northwest inland arid and semi-arid regions. Surrounded by mountains to the north and south of the city, the Yellow River flows from the southwest to the northeast and forms a beaded river valley landform with a canyon and basin; the northeastern area is low-lying, and the southwest is higher, with an average elevation between 1500 m and 1550 m. This region is affected by the typical temperate continental climate. The average annual temperature is 9.1 °C, the annual extreme high temperature and extreme low temperature are 39.1 °C and −23.1 °C, respectively, and the accumulated temperature of ≥10 °C is 3242.0 °C [33]. The annual average precipitation is 312.9 mm, with uneven seasonal distribution of precipitation mainly concentrated in the summer, and annual average evaporation is greater than 1000 mm. This study was conducted in the Poly Lingxiu mountain (36.125° N, 103.724° E) (Figure 1) of the northern mountains in Lanzhou city, Gansu Province, which is a semi-desert area. The terrain is high in the north and low in the south, sloping from the northwest to the southeast, with an elevation of 1560–2067 m, and the slope is generally >30° [33]. The main part of the soil is light sierozem with low plantation coverage and is mainly distributed in shrubs such as *Tamarix ramosissima*, *Caragana sinica*, *Reaumuria soongorica*, *Agropyron cristatum*, *Limonium arueum*, *Asterothamnus centraliasiaticus*, *Peganum multisectum*, *Artemisia barachyloba* and other herbs.

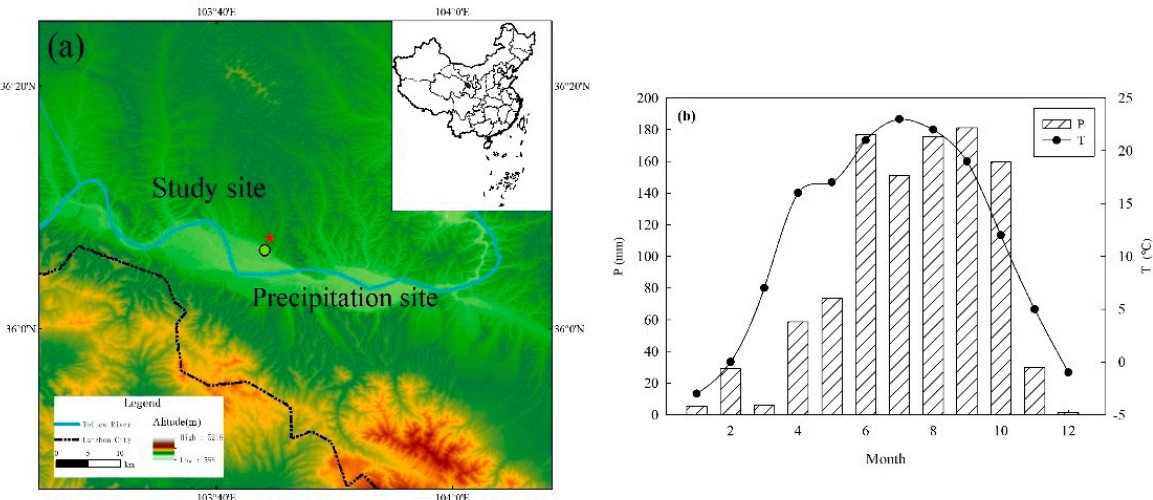

**Figure 1.** Location of the study area and sampling sites (**a**) of dominant species and the temperature and precipitation in 2019 (**b**) in the Poly Lingxiu mountain.

### 2.2. Experimental Design and Sample Collection

We attempted to complete sampling on a sunny day to avoid the influence of rainfall. In addition, in order to avoid the effect of such external environment as light and temperature, we ensured that the sampling was completed at 8:00–10:00 am on the day.

We selected two *C. korshinskii* and *R. soongorica* growing well on the gentle slope (36.125° N, 103.724° E) near Poly Lingxiu Mountain during the vegetation season (April–October) in 2019, respectively. Each plant sample was gathered from four canopy directions.

In addition, the phloem tissue of *C. korshinskii* and *R. soongorica* was removed to avoid isotopic fractionation of xylem water and contamination by isotopic enriched water [39]. All the plant samples were cut into 3–5 cm segments, immediately placed into glass vials with screw caps, sealed with polyethylene parafilm and kept frozen in a freezer (−20 °C) until isotopic analysis. We collected four parallel samples for each plant.

We excavated a 160 cm soil profile near the plant sample site, and soil samples were collected simultaneously with plant tissue sampling. Each soil column was taken at 10 cm depth intervals in the 0–100 cm range and at 30 cm intervals in the 100–160 cm layers, determined using a steel tape measure (Supplementary Materials Figure S1). Soil samples were divided into two parts: one portion was stored in a freezer until isotopic analysis and the other was used to obtain gravimetric water content of soil (SWC, %) as determined by drying at 105 °C for 24 h. Two parallel samples were collected for each respective soil layer.

Considering the convenience of rainwater sample collection, rainwater was collected in the Meteorological Park of the new campus of Northwest Normal University, which is close to the study site (2.5 km). The precipitation samples were collected immediately using a standard rain gauge after the precipitation to prevent evaporation, packed in HDPE plastic bottles, sealed with Parafilm and then stored in a freezer for isotopic analysis. A total of 30 rainwater samples were collected. The irrigation water was collected monthly, except for August and September, and was packed in HDPE plastic bottles, sealed with Parafilm and then stored in an ice locker until isotopic analysis. A total of 5 irrigation water samples was collected (Supplementary Materials Table S1).

### 2.3. Isotopic Analyses

Water in plant and soil samples was extracted using an automatic vacuum condensation extraction system (LI-2100, LICA, Beijing, China) in the laboratory of stable isotopes, College of Geography and Environmental Sciences, Northwest Normal University. The extraction efficiency was over 98%, which was sufficient to obtain unfractionated water samples [29]. Rain and extracted plant and soil water were filtered using 0.22 μm organic phase pin-type filters to eliminate impurities and organic contamination.

Isotopic measurement of water in soil, plant and precipitation were analyzed with an isotopic ratio infrared spectroscopy (IRIS) system (T-LWIA-45-EP, ABB-Los Gatos Research, CA). The analytical precision of the liquid water measured by IRIS was ±1‰ for $\delta^2$H and ±0.3‰ for $\delta^{18}$O. The measured values of $\delta^2$H and $\delta^{18}$O were expressed in thousandths relative to the Vienna Standard Mean Ocean Water (VSMOW) as follows:

$$\delta X‰ = \left(\frac{R_{\text{sample}}}{R_{\text{standard}}} - 1\right) \times 1000, \tag{1}$$

where $X$ represents $^2$H or $^{18}$O, $R_{\text{sample}}$ and $R_{\text{standard}}$ represent the molar abundance ratios ($^2$H/$^1$H and $^{18}$O/$^{16}$O, respectively) of the sample and standard (Vienna Standard Mean Ocean Water), respectively.

Extracting water from plant xylem and soils using cryogenic vacuum distillation can mix organic materials (e.g., methanol and ethanol) [40] that may affect the spectroscopy and lead to erroneous stable isotope values when analyzing with IRIS [41,42]. In order to remove the pollution of methanol and ethanol, the spectral analysis software of Los Gatos (Los Gatos Research Inc., Mountain View, CA, USA) was used to correct the soil water and plant water data [43,44].

### 2.4. Data Analyses

The study site has a dry climate, low rainfall, strong evaporation, deep soil and deeply buried groundwater (>120 m), which makes it difficult for vegetation to absorb and use water. What is more, the natural precipitation and artificial irrigation water are both infiltrated into the soil. As the main direct water sources for plants, they are crucial for green engineering of southern and northern mountains of Lanzhou city. Considering that, in this paper, soil water at different layers is only used as a potential water source. SPSS 16.0 (SPSS Inc., Chicago, Illinois, USA) was used to perform single factor analysis of variance (ANOVA) on soil water $\delta^{18}$O in each layer, and the least significant difference

method (LSD) was selected. If $P \geq 0.05$, there was no difference in $\delta^{18}O$ between two layers of soil water. According to the test results and combining the soil moisture content of each soil layer, the adjacent soil horizons were combined, and the average value of the combined soil layer soil water $\delta^{18}O$ was taken as the combined layer soil water $\delta^{18}O$. The water sources from different soil layers were combined into four larger soil layers (0–10, 10–40, 40–100 and 100–160 cm) to facilitate the subsequent analysis and comparison. Four layers were identified as follows:

1. Surface soil layer (0–10 cm): the isotopic ratios in soil water and SWC had greater variability and were vulnerable to rainfall pulse, irrigation and evaporation with season.
2. Shallow soil layer (10–40 cm): the SWC was variable with obviously differences with changing seasons.
3. Middle soil layer (40–100 cm): the isotopic ratios in soil water and SWC had gentle variations and milder monthly changes than those in the shallow soil layer.
4. Deep soil layer (100–160 cm): the isotopic ratios in soil water and SWC showed relatively stable variations.

Studies have shown that $^2H$ is prone to be fractionated more than $^{18}O$ during water absorption in arid and semi-arid areas [19]. Therefore, this paper chose $^{18}O$ in combination with Bayesian model MixSIAR to determine water sources of *C. korshinskii* and *R. soongorica* and analyzed their water strategies.

We used SPSS 16.0, SigmaPlot 12.5 (Systat Software Inc., San Jose, CA, USA) and Origin 2018 (OriginLab Corporation, Northampton, MA, USA) to organize and analyze the data and plot.

The local temperature and precipitation data in 2019 were from the weather query website (https://tianqi.911cha.com/).

## 3. Results

### 3.1. Precipitation, Irrigation Water and Isotopic Composition

The total precipitation was 1048.8 mm in 2019, mainly in June–October, and 93.1% occurred during the growing season (Figure 2). A total of 30 rainwater samples were collected during the growing season, and then their isotopic compositions were measured. The $\delta^{18}O$ ranged from −0.13 to 8.28‰ with an average of −5.61‰, and $\delta^2H$ ranged from −110.05 to 37.84‰ with a mean value of −32.69‰. The local meteoric water line (LMWL: $\delta^2H = 7.00\delta^{18}O + 3.81$, $R^2 = 0.95$, $P < 0.0001$) was calculated on the basis of precipitation isotopic data. The slope and intercept of the LMWL were smaller than those of the global meteoric water line (GMWL: $\delta^2H = 8\delta^{18}O + 10$) [45] (Figure 3), which indicates local precipitation was affected by evaporation [46], and reflects the climatic characteristics with less precipitation and greater evaporation [47,48]. The $\delta^{18}O$ of irrigation water ranged from −11.64 to −9.83‰, with an average of −10.73‰, and $\delta^2H$ of irrigation water ranged from −77.50 to −70.53‰, with a mean value of −74.41‰. The isotopic composition of irrigation water was far from that of plant xylem water, which indicates the irrigation water was not the direct water source for plants (Figure 3).

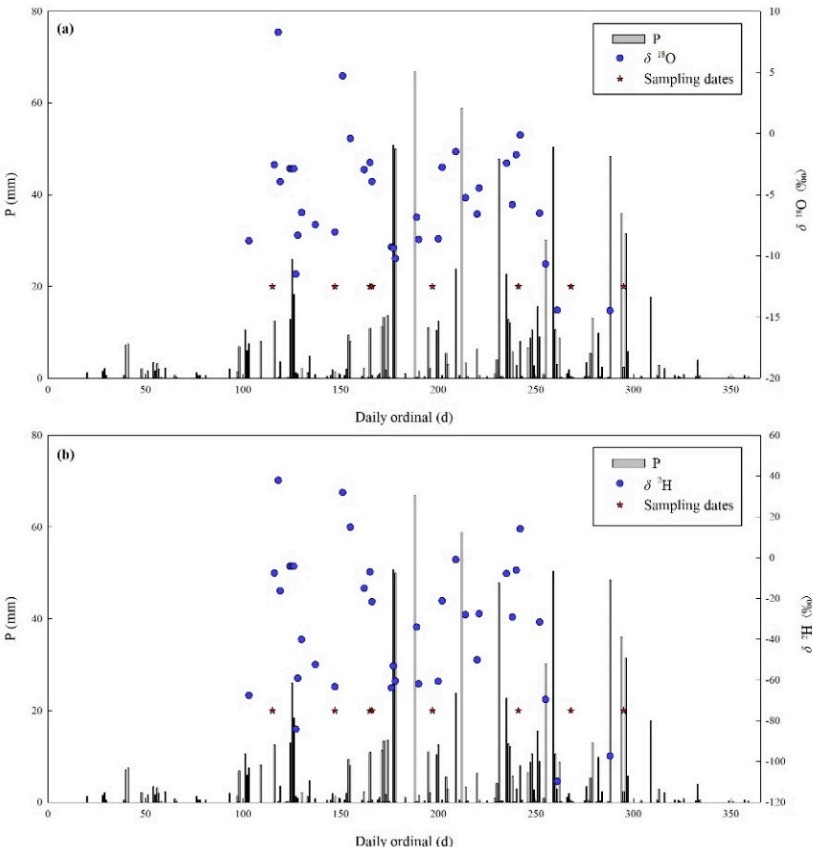

**Figure 2.** The precipitation and isotopic composition of rainwater in 2019. The blue dots in (**a**,**b**) are $\delta^{18}$O and $\delta^2$H, respectively; the red stars represent sampling dates during the growing season.

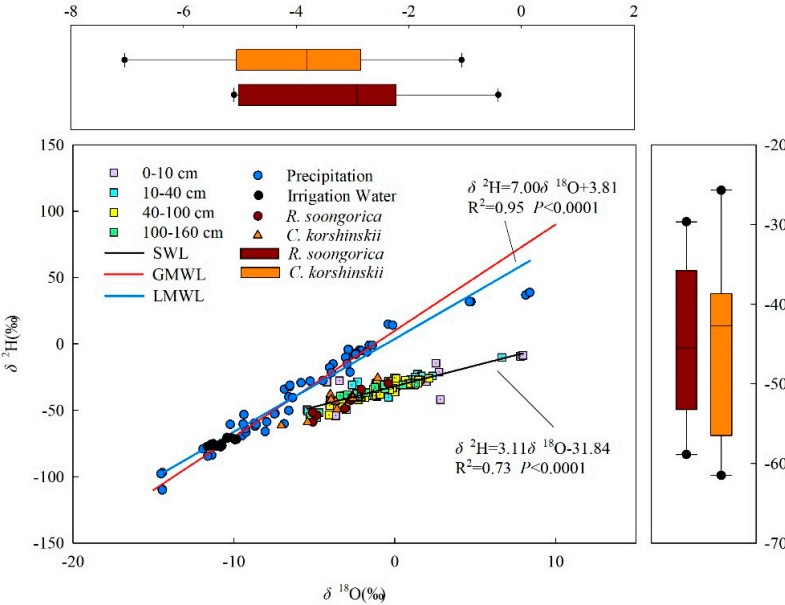

**Figure 3.** The linear regression relationship between $\delta^2$H and $\delta^{18}$O in soil water, irrigation water and rainfall during the growing season. SWL represents soil water line and is based on isotopic data of soil water ($\delta^2$H = 3.11$\delta^{18}$O − 31.84, $R^2$ = 0.73, $P$ < 0.0001). LMWL represents the local meteoric water line ($\delta^2$H = 7.00$\delta^{18}$O + 3.81, $R^2$ = 0.95, $P$ < 0.0001). GMWL is the global meteoric water line ($\delta^2$H = 8$\delta^{18}$O + 10) (Craig, 1961). GMWL are plotted for reference. The isotopic compositions of xylem water from two shrubs are shown in the figure.

### 3.2. Soil Moisture and Isotopic Composition

The soil water content and $\delta^{18}O$ variation in each of the soil depths are shown in Figures 4 and 5. The surface and shallow soils were affected by precipitation, evaporation and irrigation, etc. The water content fluctuated greatly, and seasonal differences were significant. SWC of the middle soil layer were with lower variations and milder monthly changes than that in the shallow soil layer. Except for 15 June, the moisture content of deep soil basically did not change with the depths and showed no significant differences among the sampling dates. This may be due to heavy precipitation from the night of 14 June to the next morning. Generally, the water content gradually decreased and remained stable with soil depth. The soil water line (SWL: $\delta^2H = 3.11\delta^{18}O - 31.84$, $R^2 = 0.73$, $P < 0.0001$) was drawn, based on the isotopic composition of the soil water (Figure 3), which was distributed on the bottom right of the local meteoric water line (LMWL). This may be due the soil water being affected by evaporative fractionation, and isotopes were more enriched. The $\delta^{18}O$ of soil water in the surface and shallow layer ranged from −3.67 to 7.97‰ and −5.40 to 6.67‰, respectively. The isotopic ratios in soil water manifested dramatic seasonal differences and large fluctuations, which was attributed to soil water in the surface, and the shallow layer was primary influenced by two processes, namely evaporation [48] and infiltration [49]. The $\delta^{18}O$ of soil water in the middle layer ranged from −4.08 to 2.13‰, changing smoothly. The $\delta^{18}O$ of deep soil layer ranged from −5.25 to 1.29‰, having shown little seasonal differences and relatively stable variations. In summary, the $\delta^{18}O$ of soil water decreased with depth and tended to stabilize, which indicated that the surface soil water was more susceptible to the effects of evaporation and precipitation and showed a greater difference than the deeper soil water [48], which is consistent with previous studies [50–52].

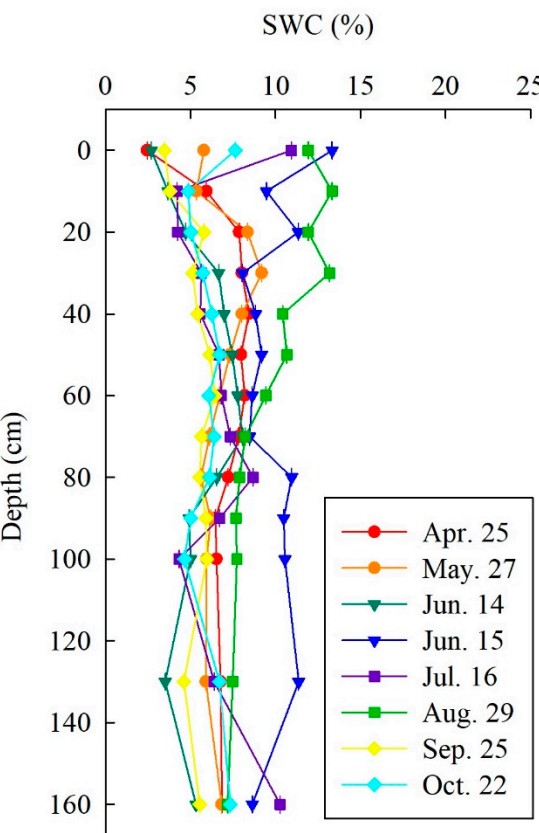

**Figure 4.** Vertical profiles (0–160 cm) of gravimetric water content of soil (SWC, %) (Mean ±S.D., N = 2).

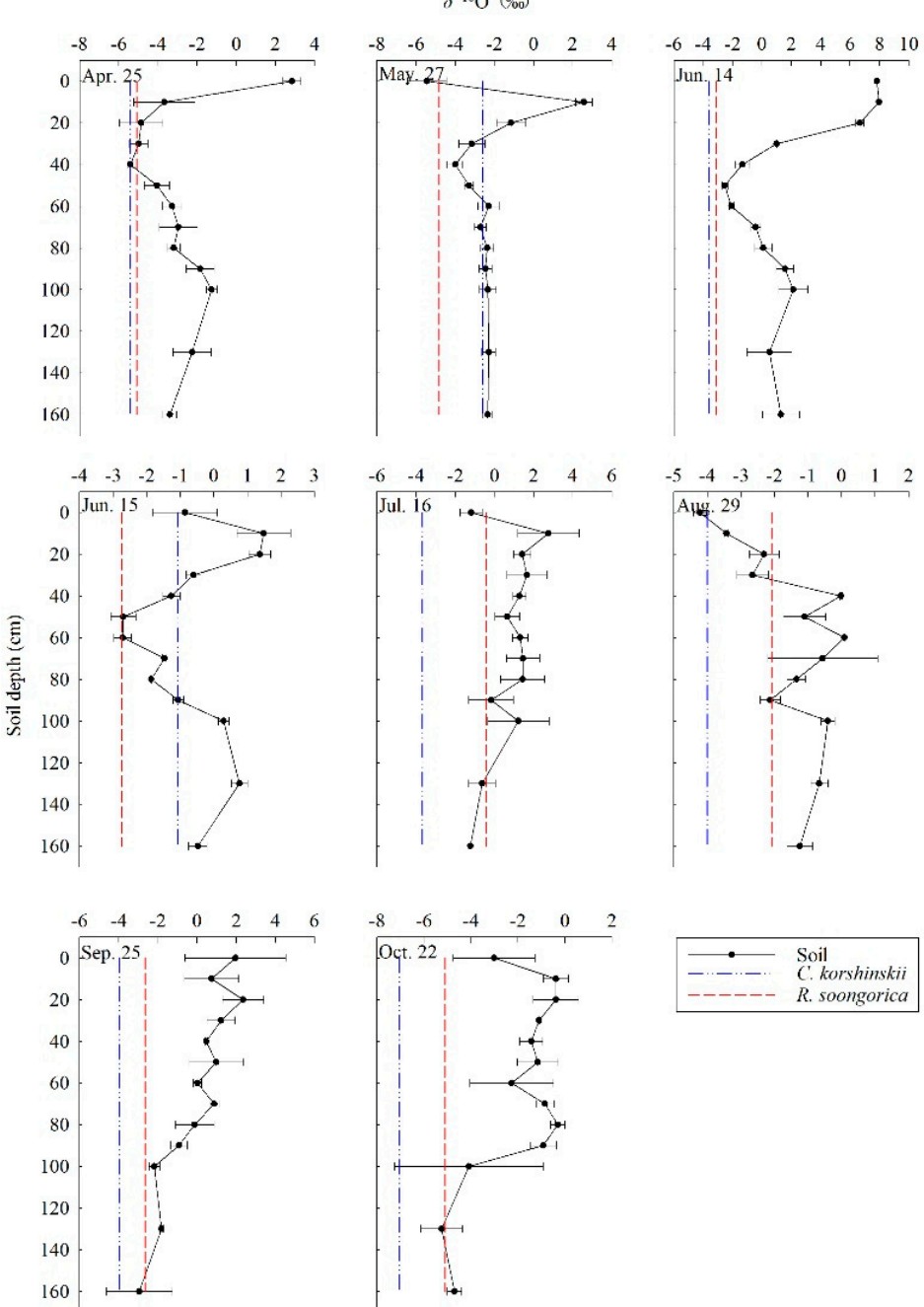

**Figure 5.** Seasonal variations of $\delta^{18}$O in soil horizons and xylem water from *C. korshinskii* and *R. soongorica*. (Mean ±S.D., N = 2).

### 3.3. Formatting of Mathematical Components

The isotopic values of plant xylem water were different among different species of the same season, and the isotopic values of xylem water of the same species in different seasons were also different. The $\delta^2$H value of *C. korshinskii* was between −61.49 and −25.69‰ with an average of −46.63‰, and $\delta^{18}$O ranged from −7.05 to −1.06‰ with an average value of −4.13‰ during the sampling periods. The $\delta^{18}$O ranged from −5.11 to −0.41‰ with an average of −3.21‰, and $\delta^2$H ranged from −58.85 to −29.64‰ with a mean value of −44.89‰ for *R. soongorica* (Figure 6). The xylem water line (XWL) was $\delta^2$H = 5.91$\delta^{18}$O − 25.93, R$^2$ = 0.86, *P* < 0.0001 and $\delta^2$H = 5.78$\delta^{18}$O − 22.77, R$^2$ = 0.78, *P* < 0.0001 for *R. soongorica* and *C. korshinskii*, respectively (Figure 7). The isotopic compositions of *C. korshinskii*

and *R. soongorica* were distributed closely to those of soil water (Figure 3), indicating that the two plants primarily obtained water from soil. There was no significant difference (P > 0.05) between the two species in averaging the xylem water isotope values over all sampling dates, implying that they absorbed water from similar soil, and there may be a certain water competition between *C. korshinskii* and *R. soongorica* (Figure 3). The isotopic composition of *R. soongorica* fluctuated in a single peak, and *C. korshinskii* changed with fluctuation, implying that *C. korshinskii* could adjust water use strategy in a flexible way during growing seasons.

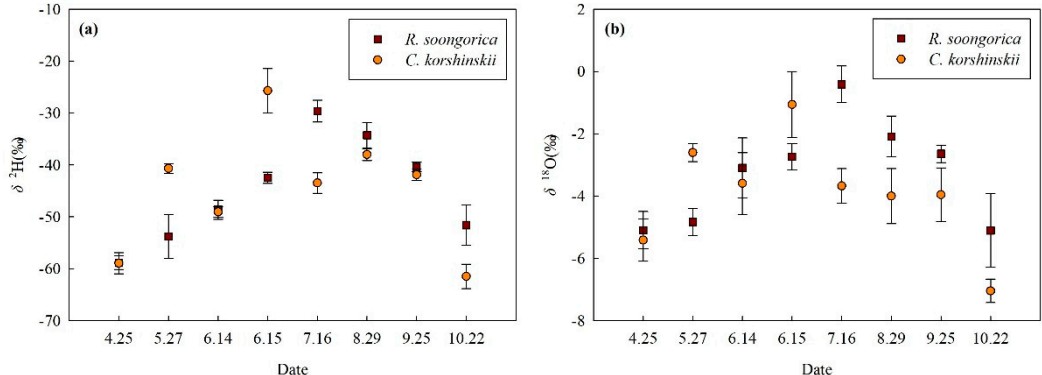

**Figure 6.** Seasonal isotopic variations of (**a**) $\delta^2$H and (**b**) $\delta^{18}$O xylem water in *C. korshinskii* and *R. soongorica*. (Mean ±S.D., N = 4).

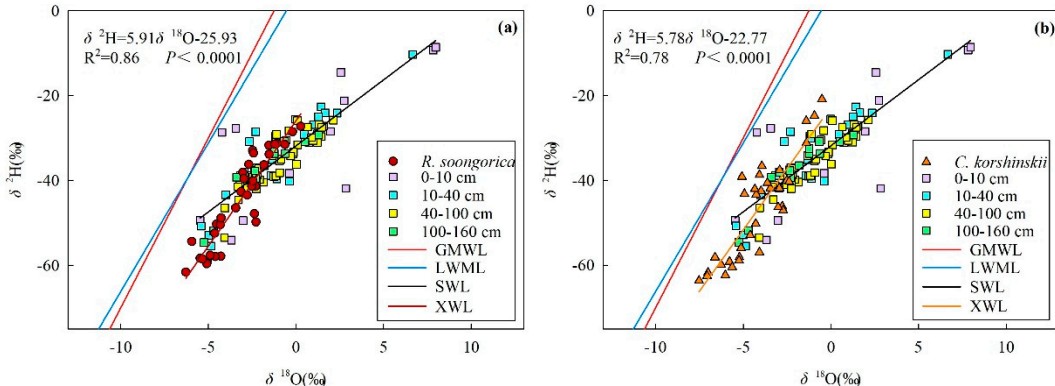

**Figure 7.** The linear regression relationship between $\delta^2$H and $\delta^{18}$O in xylem water from two species, (**a**) *R. soongorica* and (**b**) *C. korshinskii*, during the sampling periods. XWL represents xylem water line based on isotopic data of xylem water. The xylem water line (XWL) was $\delta^2$H = 5.91$\delta^{18}$O − 25.93, $R^2$ = 0.86, *P* < 0.0001 and $\delta^2$H = 5.78$\delta^{18}$O − 22.77, $R^2$ = 0.78, *P* < 0.0001 for *R. soongorica* and *C. korshinskii*, respectively. SWL, LMWL and GMWL are plotted in each panel for reference. The isotopic compositions of soil water are shown in the figure.

## 3.4. Main Water-Absorptive Horizon of Plant

According to the graphical inference method, *R. soongorica* extracted most water from the 0–40 cm and 100–160 cm soil layers during the vegetation season. On 16 July, 25 September and 22 October, the isotopic ratios of xylem water of *C. korshinskii* were far from those of the soil water, and the potential water source of xylem water could not be determined (Figure 5). Additionally, on 27 May, the isotopic ratios of xylem water of *C. korshinskii* and soil water overlapped at several soil depths, which indicated that plants may have drawn water from several soil depths. However, this method only provided the main depth of absorption and may have led to erroneous interpretation when plants extracted water from multiple soil layers simultaneously.

The MixSIAR model predicted that the ecological plasticity of water use strategies of *R. soongorica* and *C. korshinskii* was high, and the water use proportion of the two plants showed obvious seasonal differences during the sampling period (Figure 8). The water use proportion of *C. korshinskii* was less than 10% for surface soil during the vegetation season, but the utilization rate on 29 August was as high as 96.3%. The proportional contribution of shallow soil water was higher on 25 April, 27 May, and 15 June, while that of other sampling dates was less than 10%. On 27 May, 14 and 15 June and 22 October, the utilization rate of the middle soil layer was higher than 30%, and the remaining sampling dates were less than 10%. What is more, on 27 May, 15 June, 16 July and 25 September, the water use fraction for deep soil water was higher, and the other dates were less than 10%. *R. soongorica* had higher utilization of surface soil on 27 May, 16 July and 29 August. The proportional contribution of shallow soil water was higher on 25 April, 16 July and 29 August, compared to the other dates were less than 10%. On 14 and 15 June, 16 July and 29 August, it had higher utilization of middle soil, of which 14 and 15 June reached more than 98%. On 16 July, 29 August, 25 September and 22 October, it mainly absorbed deep soil water.

### 3.5. Seasonal Variations in the Proportion of Plant Water Uptake

The MixSIAR model was used to calculate the water use fraction of *R. soongorica* and *C. korshinskii* for each soil (Figure 8). On 25 April, the proportion of water from shallow soil was as high as 80% for *R. soongorica* and *C. korshinskii*, which was consistent with the high value region of soil water content. *R. soongorica* and *C. korshinskii* mainly obtained water from surface and shallow soil on 27 May, respectively. On 14 June, the middle soil layer contributed the most water up to 90% for two shrubs, which was attributed to the fact that there was no effective precipitation in the 10 days before sampling, and when the transpiration of plants was gradually enhanced, *R. soongorica* and *C. korshinskii* turned to absorb and utilize the middle layer soil water. A precipitation event occurred during the night of 14 June to the early morning of the next day, and the water content in the surface soil was high when sampling on 15 June. *C. korshinskii* can quickly respond to the precipitation pulse and flexibly change the water source. The proportions of water that *C. korshinskii* absorbed from surface, shallow, middle and deep soils were 9.40%, 12.80%, 67.10% and 10.70%, respectively. Nevertheless, the largest proportion of water (98.40%) was absorbed from deep soils for *R. soongorica* on 15 June. This showed that *C. korshinskii* could respond quickly to precipitation events, while *R. soongorica* had a slow response to precipitation pulses and chose from more conservative water sources. On 16 July, the main source was deep soil water, and the fraction for *C. korshinskii* (95.30 ± 0.04%) was higher than that of *R. soongorica* (57.30 ± 0.16%). The shallow soil contributed most water to *C. korshinskii*, which was ascribed to the high amount of precipitation in the 10 days before sampling, and the surface soil water content was abundant on 29 August. However, the fraction of *R. soongorica* for surface, shallow, middle and deep soil horizons was basically flat at 30.60%, 27.30%, 21.5% and 20.60%, respectively. After experiencing strong transpiration in the summer, the two shrubs entered the later growth period (on 25 September), and the water content of shallow soil was less and water supply was insufficient. *C. korshinskii* and *R. soongorica* both absorbed and utilized deep soil water to promote the opening of their stomata and prolong their growth and development in the dry season [53]. On 22 October, the contribution of middle soil water for *C. korshinskii* was 84.60%, and the proportion of deep soil water for *R. soongorica* was 84.40%. In general, *C. korshinskii* could flexibly switch water sources to absorb soil water in various soil layers, and *R. soongorica* gradually turned to absorb deep subsoil water and deepened the proportion during the growing season. *C. korshinskii* and *R. soongorica* both showed strong plasticity in water use strategies.

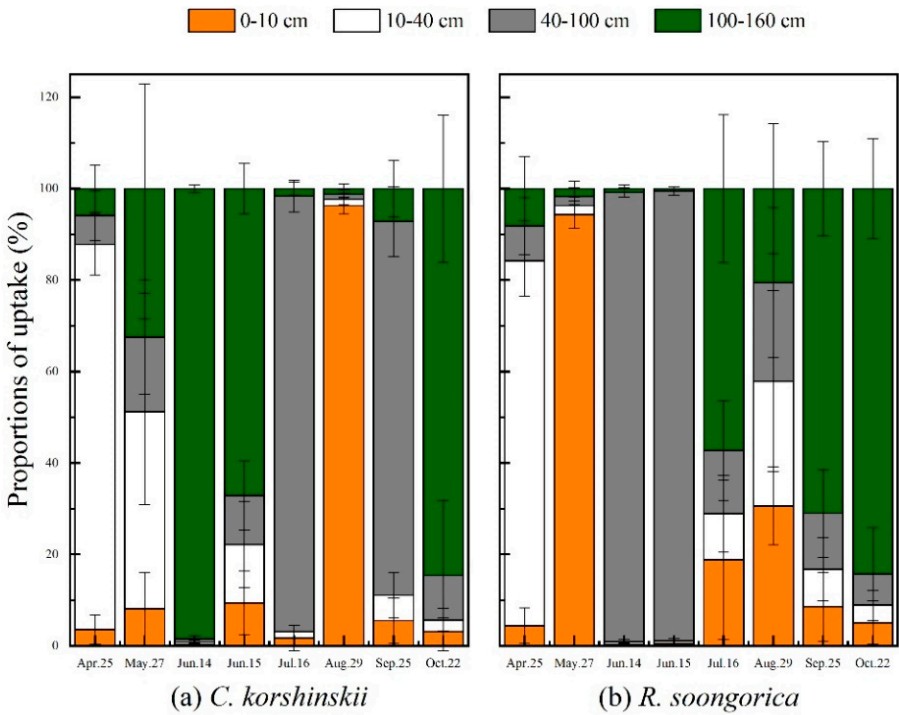

**Figure 8.** Seasonal variations in fraction of water uptake from different soil layers based on MixSIAR for (**a**) *C. korshinskii* and (**b**) *R. soongorica*. (Mean ±S.D., N = 4).

## 4. Discussion

The isotopic vertical gradient of soil water was mainly affected by the processes of evaporation and infiltration simultaneously, with the mixing of old and new rainwater [2,54]. Except 29 August, the $\delta^{18}$O in soil water became more depleted with depth. This was attributed to abundant precipitation in the 10 days before sampling, and the moisture content of surface soil was high and the $\delta^{18}$O in soil water was affected by depleted isotopes in precipitation. Compared to deeper soil profiles, the $\delta^{18}$O in surface soil water was more abundant on 25 April, 14 June and 25 September, which was attributed more to evaporation from surface soil layers [54]. Nevertheless, this trend was reversed in surface soil on the other dates during the growing season, because the rainwater mixed with depleted isotope penetrated into the surface soil through the soil matrix. Overall, the $\delta^{18}$O in soil water became more depleted with the depth and tended to stabilize, suggesting that superficial soil water was more influenced by evaporation and precipitation, and displayed greater changes than those of deep soil [48].

The *C. korshinskii* can flexibly switch water sources to absorbed soil water in various soil layers according to soil water source abundance, which is also a strategy in adapting to drought [55,56]. These results are consistent with the existence of a dimorphic root system in many species in arid and semi-arid areas, which enable them to absorb water from superficial and deeper soil layers. In addition, *C. korshinskii* can respond quickly to precipitation events, which depends on the ability of fine roots to quickly switch use patterns to cope with precipitation pulses and water shortages [57]. Besides, this flexible water use strategy will be more superior in adapting to arid environments. Zhu et al. [58] found that *C. korshinskii* could selectively similarly use soil water at different depths with growing seasons in the Ulanbu Desert. *R. soongorica* gradually turned to absorb deep soil water during the vegetation season and deepened the fraction from 0.5% to 84.4% to cope with depleted water. It showed the greater degree of plasticity in water use. This pattern may be due to the gradual increase of transpiration during the continuous growing season [2]. This is similar to the findings of other ecosystems in which woody shrubs gradually increase their water absorption depths as they grow [57]. Zhou et al. [53] found that *R. soongorica* obtained a higher percentage of water from shallow soil water

(0–100 cm) in the spring. However, during the summer and autumn, *R. soongorica* tended to use deeper soil water.

There is a certain water competition between *C. korshinskii* and *R. soongorica* in the same habitat. *C. korshinskii* can flexibly change its water use patterns to avoid the competition, which may lead to ecological complementary uses of resources and, thereby, promote species coexistence and ecosystem functions [57]. This flexible water use pattern will promote the adaptation to dry environment under the predicted increasingly frequent drought events of the future [2]. The plants start to recover and the leaves start to grow in April, which requires a lot of water supply, and the water strategies of *C. korshinskii* were similar with *R. soongorica* at the beginning of the vegetation season. Furthermore, *C. korshinskii* and *R. soongorica* face aggressive competition in water uptake at the end of the growing season. Therefore, measures such as increasing irrigation density and water volume at the beginning and end of the growing season can be used in order to alleviate competition. In the wet season with more precipitation, the competition between *C. korshinskii* and *R. soongorica* is small, and the irrigation frequency can be appropriately reduced. In this study, these different water use strategies of *C. korshinskii* and *R. soongorica* in the same habitat provided insights into ecological restoration for the southern and northern mountains of Lanzhou city.

## 5. Conclusions

We used the MixSIAR model based on stable isotopes ($\delta^{18}$O) to study the seasonal variations of water use strategies of dominant species in natural shrubs: *C. korshinskii* and *R. soongorica* on the southern and northern mountains of Lanzhou city. The results showed that the flexible water use patterns of *C. korshinskii* and its faster response to precipitation pulse compared with *Reaumuria soongorica* might promote plant species to taking full use of water and nutrients, adapting to the dry environment. However, *R. soongorica* progressively switched to suck up soil water of deep layers and deepened the proportion from 0.5% to 84.4% with the seasons, indicating a greater degree of ecological plasticity. The flexible water use strategies of *C. korshinskii* and *R. soongorica* in the same habitat reduced competition between the two shrubs for water and nutrients and enhanced adaptability to arid environments. The work presented here provides insights into ecological restoration for the southern and northern mountains of Lanzhou city.

**Supplementary Materials:** The following are available online at http://www.mdpi.com/2073-4441/12/7/1923/s1, Figure S1: Soil profile at the sampling site, Table S1: The information of irrigation at the sampling site.

**Author Contributions:** Conceptualization, M.Z. and Y.Z.; Software, Y.Z.; Formal Analysis, Y.Z.; Investigation, Y.Z., J.W., P.S., R.G., W.D., and D.Q.; Writing—Original Draft Preparation, Y.Z.; Writing—Review & Editing, Y.Z.; Project Administration, M.Z.; Funding Acquisition, M.Z. All authors have read and agreed to the published version of the manuscript.

**Funding:** This research was funded by the National Natural Science Foundation of China No. 41771035, and the Scientific Research Program of Higher Education Institutions of Gansu Province No. 2018C-02.

**Acknowledgments:** The authors greatly thank the colleagues in the Northwest Normal University for their help in sample collection and laboratory analysis.

**Conflicts of Interest:** The authors declare no conflict of interest. The funders had no role in the design of the study; in the collection, analyses, or interpretation of data; in the writing of the manuscript; or in the decision to publish the results.

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
