# Peer review of "Water Use Strategies of Dominant Species (Caragana korshinskii and Reaumuria soongorica) in Natural Shrubs Based on Stable Isotopes in the Loess Hill, China"

_water, doi:10.3390/w12071923_

Round 1

Reviewer 1 Report

The presented article discusses an interesting issue, although its scope and potential interest is regional. In terms of methodology, it employed a method thoroughly investigated earlier, applying the MixSIAR model.

In my opinion, the primary drawback of the article is lack of any information regarding irrigation, especially that according to the Authors:

Line 161-162: “What’s more, the natural precipitation and artifical irrigation water are both infiltrated into the soil”.

There is no information regarding the source of water for irrigation, its isotopic composition (such composition was analysed in detail in the case of precipitation water), the cycle of performed irrigations, etc.

Line 69-72: This fragment is more suitable for the description of the study area.

Line 72-74: In the context of the statement: “The green project in the southern and northern mountains is of great significance in improving the quality of the ecological environment in Lanzhou and promoting urban construction”...  Please elaborate on the possibilities of application of the obtained results in the discussion section.

Line 75-76 ….”but the water use pattern of typical plants remain poorly understood in this region”… Please provide a confirmation of this information in current studies – citing.

Line 103-104: “[Error! Bookmark not defined.]” Please correct throughout the paper.

Line 117-119: Please explain based on what methodical assumptions the presented system of sample collection was adopted. These are very strict guidelines with no justification. E.g. what will happen to the results in the case of a sample collected at 11:00, etc.

Line 128-129: “We excavated a 160 cm soil profile near the plant’s sample site, and soil samples were collected simultaneously with plant tissue sampling.” What was the basis for the determination of profile depth? Why is it 160 cm, and not 2 m, 2.5 m, etc.?

How deep does the root system of the described plants reach?

At what distance were the profiles performed? Did plants other than the analysed ones occur in the vicinity of the profiles? (if so, please specify)

It would be advisable to show such a profile in a photograph (I assume the Authors took such photographs).

Line 134-135: Please specify the distance between the place of precipitation measurement and the study area.

Line 360-362: The effect of shallow groundwaters on various processes is obvious (in comparison to deeper waters) due to the greatest impact of the atmosphere. This conclusion is not very insightful.

Author Response

       We would like to thank the reviewer' valuable comments about our manuscript. Those comments are all valuable and very helpful for revising and improving our paper, as well as the important guiding significance to our researches. Moreover, we have studies comments carefully and have finished the corrections. We hope it meets with approval. The main corrections in the paper and the responds to your comments at attachment.

Reviewer 2 Report

The manuscript by Zhang and co-authors is about the water use strategies in two plant species in the northern and southern mountains of Lanhzou city (China) which seems to be under a project to promote urban construction. Though the manuscript is well presented and written I think the authors should explain why they used only two species of plants between May and October, a summer period. Have these two species proven well to have water strategies elsewhere or just because they are simply dominant? If there is other type of vegetation in the study should they contribute to the water use in the studied ecosystem? What is the opinion of the authors to this? Is there any previous information on the literature regarding the studied and unstudied plant species in the northern and southern mountains of Lanhzou city? What is the effect that this water uses have on the city does it promote the growth of other plants or vegetation? Another important remark is the period of collecting why only in the summer period and not all year long? As there been any heat wave influencing the data or the sampling? Is precipitation common in the sampling period? Finally since both species behaved distinctively what do the authors foreseen in the future for this region the over dominance of C. korshinskii in comparison to R. soongorica for instance or simply the ecological restoration of the mountains.

Author Response

(The authors gave the same response as above.)

Round 2

Reviewer 2 Report

I accept the manuscript for publication in its present form.